# Loss of the liver X receptor LXRα/β in peripheral sensory neurons modifies energy expenditure

Virginie Mansuy-Aubert[1,2], Laurent Gautron[1,2], Syann Lee[1,2], Angie L Bookout[3], Christine M Kusminski[4], Kai Sun[4], Yuan Zhang[3], Philipp E Scherer[4], David J Mangelsdorf[3]*, Joel K Elmquist[1,2,3]*

[1]Division of Hypothalamic Research, University of Texas Southwestern Medical Center, Dallas, United States; [2]Department of Internal Medicine, University of Texas Southwestern Medical Center, Dallas, United States; [3]Department of Pharmacology, Howard Hughes Medical Institute, University of Texas Southwestern Medical Center, Dallas, United States; [4]Touchstone Diabetes Center, Department of Internal Medicine, University of Texas Southwestern Medical Center, Dallas, United States

**Abstract** Peripheral neural sensory mechanisms play a crucial role in metabolic regulation but less is known about the mechanisms underlying vagal sensing itself. Recently, we identified an enrichment of liver X receptor alpha and beta (LXRα/β) in the nodose ganglia of the vagus nerve. In this study, we show mice lacking LXRα/β in peripheral sensory neurons have increased energy expenditure and weight loss when fed a Western diet (WD). Our findings suggest that the ability to metabolize and sense cholesterol and/or fatty acids in peripheral neurons is an important requirement for physiological adaptations to WDs.

*For correspondence: davo. mango@utsouthwestern.edu (DJM); Joel.Elmquist@ utsouthwestern.edu (JKE)

## Introduction

Vagal afferent neurons innervate the gastrointestinal (GI) tract, pancreas, liver, and portal vein and link peripheral levels of GI nutrients as well as circulating and stored fuels (*Berthoud, 2008*). Peripheral sensory mechanisms play a crucial role in the regulation of satiation (*Schwartz, 2004*; *Bello and Moran, 2007*; *Grill, 2010*), but mechanisms underlying vagal sensing itself are still unknown. Recently, we identified an enrichment of 'lipid sensing' nuclear receptors (NRs), including liver X receptor alpha and beta (LXRα/β) in the nodose ganglia (NG) of the vagus nerve (*Liu et al., 2014*). Notably, these neurons and their processes reside outside the blood–brain barrier, enabling the potential for direct sensing of molecules released by adipose tissue or liver. LXRs are oxysterol-sensitive NRs that direct cholesterol uptake, transport, and excretion in various tissues. LXRα and LXRβ are encoded by the *Nr1h3* and *Nr1h2* genes, respectively (NR subfamily 1, group H, member 3 and 2). LXRs regulate target genes encoding for ATP-binding cassette proteins, and apolipoproteins (*Repa et al., 2000a; Venkateswaran et al., 2000*; *Chawla et al., 2001*; *Bradley et al., 2007*; *Hong et al., 2012*). Ligand activation of LXRs also stimulates de novo lipogenesis of triglycerides in liver via sterol regulatory element-binding protein 1c (*Repa et al., 2000b; Zhang et al., 2012*). Hepatic LXRα/β regulate whole lipid and glucose homeostasis, and LXRα/β in macrophages regulate inflammation (*Chawla et al., 2001*; *Rong et al., 2013*; *A-Gonzalez et al., 2013*; *Zelcer and Tontonoz, 2006*). In the central nervous system, they regulate local inflammation, differentiation, and neuron survival by orchestrating cholesterol uptake and efflux (*Wang et al., 2002*).

**eLife digest** The vagus nerves run from the brainstem to the heart and the digestive system and help to control several processes including digestion and heart rate. Because of their role in regulating food intake, these nerves are attractive targets for scientists hoping to develop treatments for obesity.

There are two types of fat tissue found in mammals: white fat, which is used as an energy store and makes up most of the extra fat seen in obese individuals; and brown fat, which can generate body heat. The vagus nerves monitor fat and cholesterol levels in the body via receptor proteins that respond to messages sent from the fat tissues and the liver. Previous research unexpectedly found that mice genetically engineered to lack these receptor proteins—called LXRα and LXRβ—do not become obese even when fed a high-fat, high-cholesterol diet that would make normal mice gain excessive weight.

Mansuy-Aubert et al. have now investigated in more detail why mice without these receptor proteins are resistant to obesity. When fed a high-fat, high-cholesterol diet, mice that lacked the LXRα and LXRβ receptors in sensory neurons had higher cholesterol levels in their nerve cells than normal mice on the same diet. Mice lacking these receptors also burned more energy and gained less weight than normal mice.

Next, Mansuy-Aubert et al. examined fat tissue from both types of mice. This revealed that the heat-generating brown fat was more active in mice lacking the LXRα and LXRβ receptors. Some of the white fat in these mice had also become more like brown fat, allowing the mice to burn more energy and so gain less weight.

In many Western countries, many people also eat a diet that is high in fat and cholesterol. This raises the possibility that drugs that block the LXRα and LXRβ receptors in sensory neurons in humans could help to treat or prevent obesity, although further work will be needed to investigate this.

Further studies in LXR null mice revealed the rather surprising finding that these mice were resistant to obesity when challenged with a diet containing both high fat and cholesterol (*Kalaany et al., 2005*). This study showed that the LXR$^{-/-}$ response was due to abnormal energy dissipation resulting in part from ectopic expression of uncoupling proteins in white adipose. Here, we show that Western diet (WD)-fed mice that lack LXRα/β in sensory neurons of the NG have altered neuronal cholesterol content and increased white adipose tissue browning, leading to changes in energy expenditure and body weight. This unexpected function of LXRs in vagal sensory neurons provides a plausible mechanism that may in part explain the role LXRs on metabolism in response to a diet containing fat and cholesterol.

## Results

### The absence of LXRs in NAV1.8 expressing neurons disrupts *Abca1* regulation and increases NG neuronal cholesterol content

We first assessed the effects on LXR-target gene expression following pharmacological administration of LXR agonists. Canonical LXR target genes, including ATP binding cassette protein A1 (*Abca1*) and SREBP-1c (*Srebf1*), were up-regulated in the NG of mice treated with LXR agonists (*Figure 1A*). These results agree with reports of expression of *Abca1* in the central nervous system (CNS) and its up-regulation in cultured neurons and sciatic nerves following LXR agonist treatment (*Fukumoto et al., 2002*; *Cermenati et al., 2010*). Previous data have also shown SREBP-1c to be expressed in Schwann cells and CNS neurons, and LXR agonist stimulated *Srebp1f* expression has also been reported in various tissues (*Cermenati et al., 2010*). Carbohydrate-responsive element-binding protein (*Chrebp*) expression remained unchanged in response to LXR agonist (*Figure 1A*).

These results suggest that *Abca1* and *Srepb1f* are targets of LXR in the NG. To confirm these findings, we established NG organotypic cultures from mice lacking LXRα and LXRβ in vagal sensory neurons expressing the sodium ion channel NAV1.8 encoded by the *Scn10a* gene. The resulting LXRs$^{Nav}$ mice were generated by breeding double floxed *Nr1h3* and *Nr1h2* mice (LXRs$^{fl/fl}$ mice) with the Nav1.8::Cre mice, which selectively expresses Cre-recombinase in peripheral sensory neurons

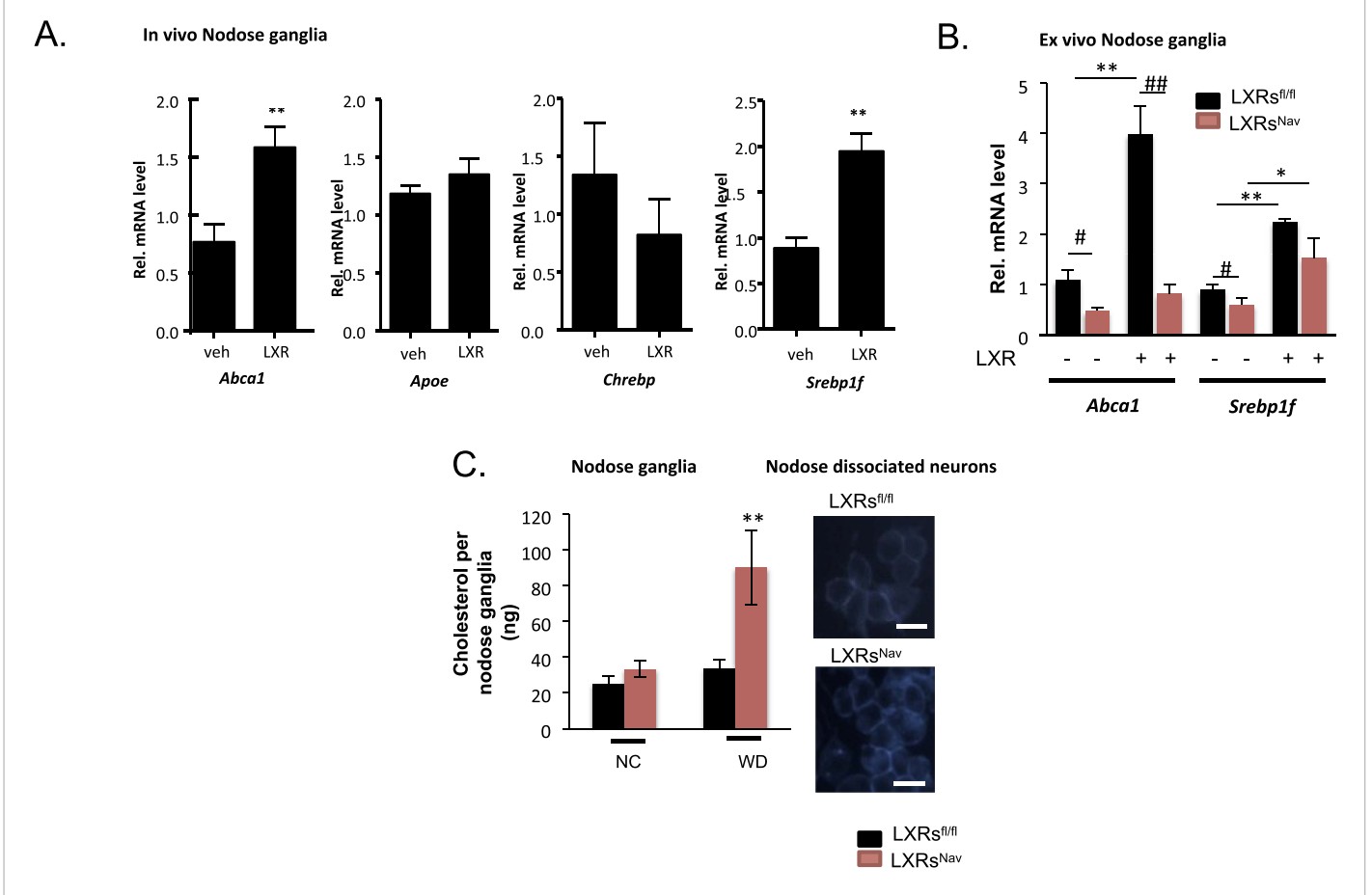

**Figure 1**. LXRs signaling regulates cholesterol metabolism in nodose ganglia neurons. (**A**) Regulation of selected target genes in nodose ganglia (NG) following liver X receptor (LXR) agonist treatment in vivo (left panel) n = 5–8 per group. **p < 0.005. (**B**) NG organotypic slices were prepared from LXRs[Nav] or LXRs[fl/fl] treated with GW3965 (5 μM) or vehicle for 4 hr. Quantitative PCR (qPCR) data are expressed as average fold-change relative to vehicle ± S.E.M., n = 3 independent experiments. # and *indicates p < 0.5, **p < 0.005. (**C**) Quantification of total cholesterol. Values were expressed as ng of cholesterol per NG, n = 6. ## and **p < 0.001. (right panel). Neurons isolated from LXRs[Nav] or control mice NG were subjected to Filipin staining (representative images of staining from 3 individual mice).

The following figure supplement is available for figure 1:

**Figure supplement 1**. Ablation of LXR in the NAV1.8 positive neurons.

(*Stirling et al., 2005*; *Gautron et al., 2011*). As expected, LXRα/β were deleted in the majority of NG neurons in LXRs[Nav] mice compared to LXRs[fl/fl] littermate controls (*Figure 1—figure supplement 1*). No difference in LXRs expression was observed in liver, white adipose tissue, brown adipose tissues (BATs), and muscle (*Figure 1—figure supplement 1*).

In LXRs[fl/fl] controls, *Abca1* and *Srebp1f* mRNA levels increased significantly after agonist treatment. This stimulation was fully (*Abca1*) or partially (*Srebp1f*) blunted in NG from LXRs[Nav] littermates (*Figure 1B*). Collectively, these results suggest that in the NG, the regulation by LXRs of *Abca1* is restricted to sensory neurons, but that its action on *Srebp1f* likely occurs in multiple cell types.

Interestingly, cholesterol assays on whole NG taken from WD (42% fat, 0.2% cholesterol)-fed LXRs[Nav] knockout mice showed a 60% increase in total cholesterol compared to littermate controls (*Figure 1C*). An increase of intracellular cholesterol level was also observed in LXRs[Nav] dispersed neurons (*Figure 1C*).

Collectively, our results suggest that in a WD setting, LXRs regulate NG sensory neuron cholesterol levels though ABCA1-dependent signaling. A role for ABCA1 in cholesterol, lipid distribution has previously been extensively observed in the brain (*Tam et al., 2006*; *Yang et al., 2006*; *Kruit et al., 2010*).

In addition, LXR agonist or ABCA1 over-expression has been shown to alter cholesterol content in degenerating neurons or cancer cells (*Smith and Land, 2012*). Our results demonstrate that in sensory neurons, NG cholesterol metabolism is also regulated via LXR.

## Resistance to diet induced obesity in mice lacking *LXRα/β* in *Nav1.8*-expressing neurons

To measure the physiological impact of LXRs loss in peripheral sensory neurons expressing Nav1.8, LXRs$^{fl/fl}$, LXRs$^{Nav}$, Nav1.8::Cre were maintained for 16 weeks on a standard rodent chow diet (normal chow, NC, 4% fat) or WD. All mice had similar body weights at weaning. However, LXRs$^{Nav}$ mice weighed significantly less than controls after 11 weeks of NC (*Figure 2A*). Similarly, LXRs$^{Nav}$ mice fed WD were resistant to diet-induced obesity. NMR analysis revealed that LXRs$^{fl/fl}$ mice had twofold more body fat after 10 weeks of WD than LXRs$^{Nav}$ mice (*Figure 2B*). However, the differences in body fat do not completely reflect the body weight difference, despite no difference in lean mass (*Figure 2B*). WD-fed LXRs$^{Nav}$ mice had higher energy expenditure than littermate controls (*Figure 2C,D*). However, no differences in food intake were noted in metabolic chambers. Furthermore, no important changes were found in plasma glucose, serum cholesterol, free fatty acids and liver triglycerides or cholesterol (*Figure 2—figure supplement 1A-F*). A decrease in adiposity in the setting of nutrient excess is sometimes due to ectopic lipid deposition in liver. However, histological examination using Hematoxylin and Eosin staining (H&E staining) showed that when fed WD, both LXRs$^{fl}$ and LXRs$^{Nav}$ mice developed hepatosteatosis and increase in hepatic lipid droplets (*Figure 2—figure supplement 1G*). Our results suggest that LXRs$^{Nav}$ mice have an exacerbated response to the WD with increased diet-induced thermogenesis. This suggests that LXRs may regulate the whole-body energy expenditure according to the amount of fat or cholesterol present in the diet.

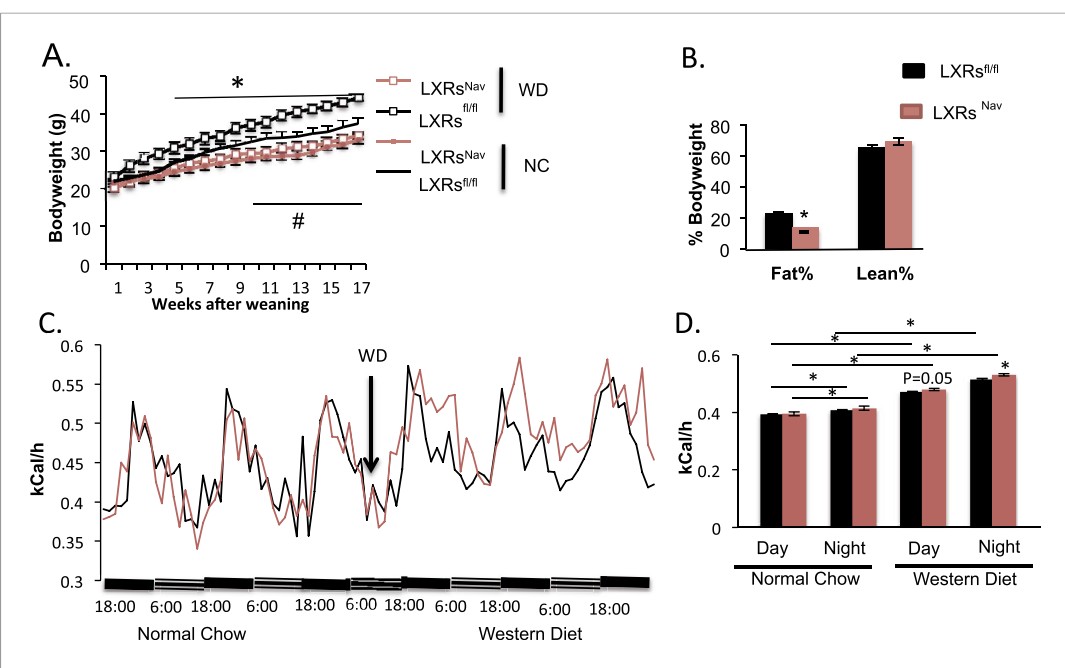

**Figure 2**. Ablation of LXRs from NAV1.8 expressing neurons exacerbates high-fat diet induced thermogenesis. (**A**) Body weight of LXRs$^{Nav}$ mice and littermate controls on chow and Western Diet (WD) as followed over time (n = 12 per genotype) * for WD, # for normal chow (NC). (**B**) Adipose tissue as a percentage of total body weight in mice fed WD for 9 weeks, measured by NMR (n = 6). (**C,D**) Energy expenditure in weight-matched mice. (**C**) Calorimetry trace before and after a switch from NC to WD. (**D**) Energy expenditure during light and dark cycles before and after a switch from NC to WD. Error bars show S.E.M. *indicates p < 0.05.

The following figure supplement is available for figure 2:

**Figure supplement 1**. Blood chemistry and lipid levels of LXRs$^{Nav}$ mice.

## The loss of LXRα/β in NAV1.8 positive neurons attenuates lipid accumulation in BAT, promotes browning in subcutaneous fat, and enhances mitochondrial respiration in skeletal muscle

Expectedly, we found that WD-fed control mice accumulated large lipid droplets in their BAT. Notably, this accumulation was considerably attenuated in LXRs[Nav] mice (*Figure 3A*). To evaluate the BAT activity, markers for mitochondrial metabolism and thermogenesis regulation in adipose tissue were assessed by Western blot or real-time PCR. Uncoupling protein 1 (*Ucp1*) and peroxisome proliferator-activated receptor gamma coactivator 1-alpha (*Pgc1α*) were higher in LXRs[Nav] mice BAT (*Figure 3B,C*). A more striking threefold UCP1 increase was observed in LXRs[Nav] mice subcutaneous fat as assessed by immunohistochemistry (IHC) and whole fat pad Western blot (*Figure 3D,E*). These results suggest LXRs in sensory neurons may be important for BAT

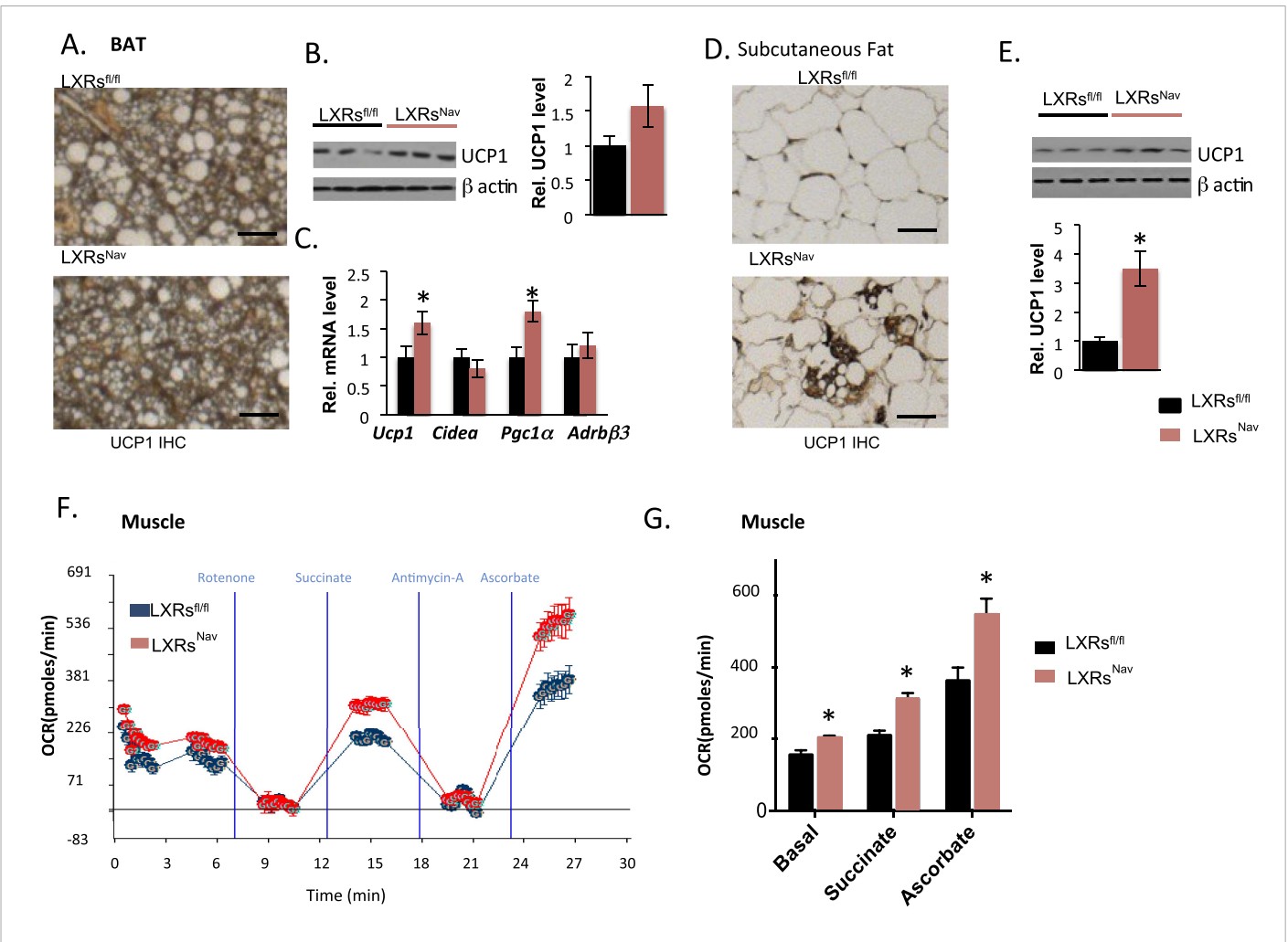

**Figure 3**. Adipose tissue and muscle reprogramming in LXRs[Nav] mice vs control mice. (**A**) Immunohistochemistry for UCP1 in BAT. (**B**) UCP1 Western blot on whole BAT pads (upper panel, n = 4) UCP1 molecular weight (MW) is 33 kDa, Actin served as the loading control, MW 42 kDa, the graph represents blot quantification. (**C**) mRNA levels in the BAT of LXR ablated mice vs control mice (lower panel, n = 4). *indicates p < 0.05. (**D**) UCP1 staining and (**E**) Western-blot analysis of UCP1 protein levels in whole, individual dorsal subcutaneous fat pads. Actin served as the loading control. The signal is quantified in the graph below (n = 4) Scale bar = 100 μm. (**F,G**) Oxygen-consumption rates were determined using the XF24 Extracellular Flux Analyzer following the manufacturers' protocols (n = 3). *indicates p < 0.01.

The following figure supplement is available for figure 3:

**Figure supplement 1**. Electron-flow experiments.

activity regulation and subcutaneous white fat conversion to a 'beige' phenotype in response to WD.

To establish whether sensory neuron-specific deletion of LXRs alters skeletal muscle mitochondrial respiration, we examined mitochondrial electron transport chain activity by performing mitochondrial electron-flow (EF) and electron-coupling (EC) experiments to assess oxygen-consumption rates (OCRs). During EF analyses, we observed that skeletal muscle mitochondria derived from LXRs^Nav mice exhibit markedly higher OCRs in response to the substrates pyruvate, malate, succinate, and ascorbate (*Figure 3F,G*), when compared from control mice skeletal muscle mitochondria. Furthermore, EC experiments to gage mitochondrial coupling and integrity revealed no defects in skeletal muscle mitochondrial function in either genotype (*Figure 3—figure supplement 1*). Taken together, these data indicate that deletion of LXR specifically in peripheral sensory neurons enhances skeletal muscle mitochondrial oxidative respiration.

## The loss of LXRα/β in NAV1.8 expressing neurons modifies NG gene expression

To gain more insights into the signaling downstream of LXR in NG sensory neurons, we surveyed genes important for vagal neuronal function, including: cholecystokinin A and B receptor (*Cckar* and *Cckbr*), which regulate vagal nerve activity and feeding; neuregulin 1 (*Nrg1*), involved in axon/Schwann cell communication and sensory nerve structure (*Gambarotta et al., 2013*; *Stassart et al., 2013*); and α, β, and γ synuclein (*Syna, Synb, Syng*), which are involved in intraneuronal trafficking/cell–cell communication and known to be LXR targets in the brain (*Golovko et al., 2009*). *Nrg1* and *Syng* were decreased twofold in both chow and WD-fed LXRs^Nav mice, interestingly *Syna* was only significantly up-regulated in control mice in response to WD (*Figure 4A*). Interestingly, in recent reports, γ synuclein has been linked to metabolism (*Oort et al., 2008*; *Golovko et al., 2009*; *Millership et al., 2012*). γ synuclein is a protein that modulates synaptic trafficking in neurons but also lipid droplets generating intracellular fatty acids. Notably, the γ synuclein whole body knockout is protected against diet-induced obesity (*Oort et al., 2008*; *Millership et al., 2012*).

Starvation reduces triglyceride levels in serum, but increases circulating fatty acids, which are rapidly taken up by the liver. To study the ability of NG to acutely respond to nutrient changes (including triglycerides and fatty acids), we studied the expression of LXR targets in the NG of fed or fasted mice. We asked whether fasting-induced increases in fatty acid availability modified NG gene expression. *Abca1*, *Srepb1f*, and *Syng* mRNA levels were significantly changed in LXRs^fl/fl mice fasted 20 hr (*Figure 4B*). Notably, the fasting-induced increase in *Abca1* and *Syng* was blunted in LXRs^Nav mice (*Figure 4B*). We also exposed NG cultures to serum obtained from mice fed or fasted for 20 hr. These data suggest that NG neurons may sense circulating secreted starvation cues and respond by regulating unique LXR-dependent genes. Despite its well-documented role in regulating the transcription of genes crucial for lipid synthesis and storage upon cholesterol sensing, little is known about how LXRs function in peripheral neurons and further investigation is necessary to completely understand the role of these NRs in the NG neurons.

Our study suggests that LXRs in vagal sensory neurons potentially regulate vagal synaptic transmission to ultimately affect the gating of information to adipose tissues and muscle. Since WAT, BAT, and muscle receive innervation from sympathetic neurons, we suspect that the increased sympathetic tone (secondary to altered input from the vagal sensory neuron activity) may underlie the increased energy expenditure observed in LXRs^Nav mice.

Interestingly, Kalaany et al., also described LXR null mice as resistant to obesity when challenged with a Western-style diet containing high fat and cholesterol. This phenotype was surprisingly independent of SREBP-1c and due to a net increase in the energy utilization in white adipose and muscle (*Kalaany et al., 2005*). Our study of LXR function in NG neurons is consistent with this previous report and provides a partial explanation of how LXRs can affect the whole body thermogenesis via sensory neurons.

*Diano et al., (2011)*, previously showed induction of LXRs in the hypothalamus in response to high-fat feeding. This finding suggests that LXRs may also regulate energy homeostasis beyond the nodose nucleus neurons.

Based on the aforementioned findings, we postulate that the LXR pathway may mediate certain aspects of lipid sensing in peripheral sensory neurons. Our findings also suggest that the ability to metabolize and sense cholesterol and/or fatty acids in peripheral neurons may be an important

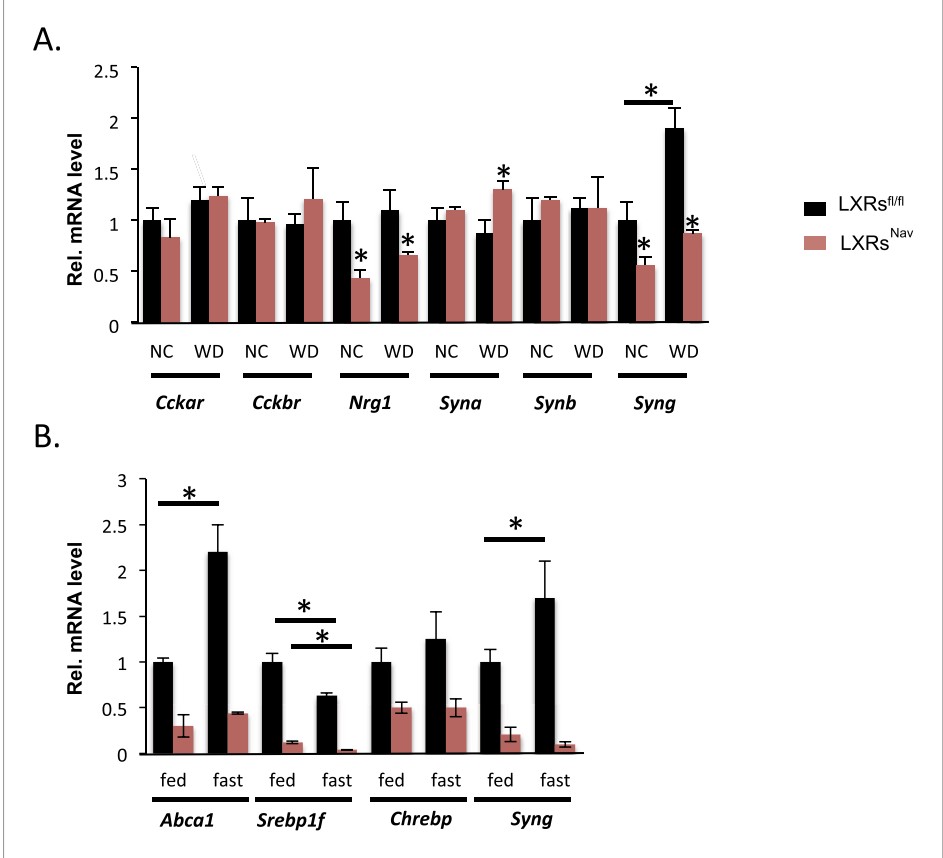

**Figure 4**. NG gene expression in LXRs$^{Nav}$ mice vs control mice. (**A**) Gene expression in NG from LXRs$^{Nav}$ and control mice fed with NC or WD. (**B**) Gene expression in NG from LXRs$^{Nav}$ and control mice fasted 20 hr or fed. All genes show a significant difference between both genotypes. *indicates $p < 0.05$ (n = 6–10).

requirement for physiological adaptations to high-fat/high-cholesterol (Western) diets, such including thermogenesis leading to changes in key metabolic processes.

## Materials and methods

### Animals

All 'Materials and methods' were approved by the Institutional Animal Care and Use Committee at UT Southwestern Medical Center. All mice were housed in a temperature-controlled room with a 12-hr light/dark cycle in the animal facility of University of Texas Southwestern Medical Center.

Mice were fed with either NC (#2916, Harlan-Teklad, Madison, WI; 4.25% kcal from fat) or WD (#88137, Harlan Teklad, cholesterol (0.2% total cholesterol), fat (42% kcal from fat), high in saturated fatty acids (>60% of total fatty acids)). LXRs$^{fl/fl}$ mice (floxed *Nr1h3* and Nr1h2 *genes*) on a mixed C57BL6 and 129SV background were bred and kept as a mixed background in a closed colony in UTSW germ-free facility. LXRs$^{fl/fl}$ mice were then backcrossed to C57BL/6J mice for six generations prior to experiments. Transgenic mice (C57/BL6 background) carrying *Cre* recombinase driven by a Scn10a promoter (called Nav1.8::Cre mice) were bred with back-crossed LXRs$^{fl/fl}$ mice. LXRs$^{fl/fl}$ mice were bred with LXRs$^{fl/fl}$ Nav$^{+/-}$ to generate cohorts of littermate LXRs$^{Nav}$ and LXRs$^{fl}$ that were used and compared in at least three cohorts for each experiment.

We have generated cohort of Nav1.8-Cre/ tdTomato reporter mice to examine the hypothalamus. Our current findings show that tdTomato Nav1.8 expression is absent from the entire hypothalamus (Data not shown).

## Body weight and composition
Body weight was monitored weekly from weaning (at 4 weeks of age) to 28 weeks of age. In the WD studies, mice were maintained on NC until 6 weeks old before being fed WD.

## Metabolic cage studies
Food intake, meal patterns, energy expenditure, and physical activity were continuously monitored using a combined indirect calorimetry system (TSE Systems GmbH, Bad Homburg, Germany). Experimental animals (11-week-old) were acclimated in the metabolic chambers for 5 days before data collection. Mice were initially maintained on NC during the acclimation period and the first two days of data collection and then fed WD for the next three days. $O_2$ consumption and $CO_2$ production were measured to determine the energy expenditure. In addition, physical activity was measured using a multi-dimensional infrared light bean system.

## In vivo agonist treatment
C57Bl/6 males were treated with vehicle (1% methylcellulose) and LXR agonist (10 milligram/kg body weight (mpk) GW3965), by oral gavage at 14 hr and 2 hr prior to sacrifice. NG were rapidly dissected and frozen in liquid nitrogen.

## Metabolic cage studies
Metabolic parameters were continuously monitored using a combined indirect calorimetry system (TSE Systems GmbH) as described in the supplemental methods.

## Data analysis
The data were represented as mean ± S.E.M. Statistical analyses were performed using GraphPad PRISM version 6.0. Single comparisons were made using 1- or 2-tailed *t* tests, as appropriate, and multiple comparisons were performed using 1-way analysis of variance (ANOVA) followed by Dunnett's post hoc test. For repeated measures, 2-way repeated-measures ANOVA was performed, with Bonferroni post hoc tests. A p value less than 0.05 was considered significant.

## Quantitative PCR
Real-time quantitative PCR (qPCR) gene expression analysis was performed using inventoried TaqMan Gene Expression Assays (Applied Biosystems). 18 s was used as normalizer. TaqMan probes used for qPCR include *18 s* (ABI, Hs99999901_s1), *Adrb3* (ABI, Mm02601819_g1), C*ckar* (ABI, Mm00438060_m1), *Cckbr* (ABI, Mm00432329_m1), *Ucp1* (ABI, Mm01244861_m1), *Pgc1a* (01016719_m1), *Nrg1* (Mm01212130_m1), S*yna* (Mm01188700_m1), S*ynb* (Mm00504325_m1), S*ynb* Mm00488345_m1), *Abca1* (Mm00442646_m1), Two Sybr green-based primer sets located in the first exons of the floxed *Nr1h3* and Nr1h2 *genes* were used to specifically detect the truncated form of LXRα and β.

## NG organotypic culture
Mouse pups between 8 and 11 day old were decapitated, and the NG were quickly removed and cultured in chilled Gey's Balanced Salt Solution (Invitrogen) enriched with glucose (0.5%) and KCl (30 mM). The NG were then placed on Millicell-CM filters (Millipore; pore size 0.4 μm) and then maintained at the air-media interface in minimum essential medium (Invitrogen) supplemented with heat-inactivated horse serum (25%, Invitrogen), glucose (32 mM), and GlutaMAX (2 mM, Invitrogen). Cultures were typically maintained for 10 days in standard medium, which was replaced three times a week. After an overnight incubation in low serum, (1.5%) MEM supplemented with GlutaMAX (2 mM), slices were stimulated with vehicle, 5 μM GW3965 for 4 hr. RNA was harvested using Acturus PicoPure RNA Extraction kit (Applied Biosystems).

## Histology
For IHC, sections were deparaffinized and the wax at the surface was removed with xylenes. After antigen retrieval and blockage of endogenous peroxidase activity, sections were stained with primary

antibodies against UCP-1 (Cat# ab10983, Abcam) followed by biotinylated secondary antibodies (anti-rabbit; Dako, Glostrup, Denmark). Secondary antibodies were detected using a DAB chromogen A kit (Dako) following the manufacturer's protocol. The slides were also counterstained with Hematoxylin. Filipin staining for unesterified cholesterol was performed according to manufacturer's instructions (FilipinIII cholesterol detection, Abcam).

### Glucose tolerance tests

After measuring the fasting glucose levels, mice were given an i.p. dose of glucose (1.5–2 g/kg body weight). Blood glucose levels were then monitored using an AlphaTrak glucometer (Abbott Laboratories, North Chicago, IL) designed for use in rodents.

### Cholesterol quantitative measurement

Adult tissue was resuspended in 1× lysis buffer placed on ice 30 min and homogenized in 2 ml tubes, glass bead-containing tubes. Samples were then directly used to quantify total cholesterol. The reaction was performed in 96-well plates by adding Amplex Red reagent, horseradish peroxidase, cholesterol oxidase, and cholesterol esterase (Amplex Red Cholesterol Assay Kit; Life technologies). The reactions were incubated for 30 min at 37°C. Results presented here were obtained from individual mice (n = 6).

### Isolation of mitochondria from skeletal muscle tissues

To isolate mitochondria, skeletal muscle tissues were homogenized using a motorized Dounce homogenizer in ice-cold MSHE buffer (70 mM sucrose, 210 mM mannitol, 5 mM HEPES, 1 mM EDTA) containing 0.5% FA-free Bovine Serum Albumin (BSA). Homogenates then underwent low centrifugation (800×$g$ for 10 min) to remove nuclei and cell debris, followed by high centrifugation (8000×$g$ for 10 min) to obtain the mitochondrial pellet, which was washed once in ice-cold MSHE buffer and was resuspended in a minimal amount of MSHE buffer prior to determination of protein concentrations using a BCA assay (Pierce).

### Mitochondrial experiments

OCRs were determined using the XF24 Extracellular Flux Analyzer (Seahorse Bioscience, MA) following the manufacturers' protocols. For the EF experiments, isolated skeletal muscle mitochondria were seeded at 10 µg of protein per well in XF24 V7 cell-culture microplates (Seahorse Bioscience), then pelleted by centrifugation (2000×$g$ for 20 min at 4°C) in 1× MAS buffer (70 mM sucrose, 220 mM mannitol, 10 mM $KH_2PO_4$, 5 mM $MgCl_2$, 2 mM 4-(2-hydroxyethyl)-1-piperazineethanesulfonic acid (HEPES), 1 mM ethylene glycol tetraacetic acid (EGTA) in 0.2% FA-free BSA; pH 7.2) supplemented with 10 mM pyruvate, 10 mM malate, and 4 µM carbonyl cyanide 4-(trifluoromethoxy)phenyl-hydrazone (FCCP) (for EF experiments), with a final volume of 500 µl per well. For EC experiments, 1× MAS buffer was supplemented with 10 mM succinate and 2 µM rotenone. The XF24 plate was then transferred to a temperature-controlled (37°C) Seahorse analyzer and subjected to a 10-min equilibration period and 2 assay cycles to measure the basal rate, comprising a 30-s mix, and a 3-min measure period each; and compounds were added by automatic pneumatic injection followed by a single assay cycle after each; comprising a 30-s mix and 3-min measure period. For EF experiments, OCR measurements were obtained following sequential additions of rotenone (2 µM final concentration), succinate (10 mM), antimycin A (4 µM), and ascorbate (10 mM) (the latter containing 1 mM N,N,N′,N′-tetramethyl-p-phenylenediamine [TMPD]). For EC experiments, OCR measurements were obtained post sequential additions of ADP (4 mM), oligomycin (2 µM), FCCP (4 µM), and antimycin-A (2 µM). OCR measurements were recorded at set interval time-points. All compounds and materials above were obtained from Sigma–Aldrich.

## Acknowledgements

We thank the UTSW Mouse Metabolic Phenotyping Core (NIH PL1 DK081182, UL1 RR024923). This work was supported by NIH grants R01DK088423 and R37DK053301 (to JKE), R01DK55758 and R01DK099110 (to PES), P01DK088761 (to PES and JKE), NURSA grant U19DK062434 (to JKE and DJM), GM007062 (to ALB); the Robert A. Welch Foundation (grant I-1275 to DJM); and the Howard Hughes Medical Institute (DJM).

# Additional information

## Competing interests

JKE: Reviewing editor, *eLife*. The other authors declare that no competing interests exist.

## Funding

| Funder | Grant reference | Author |
| --- | --- | --- |
| National Institutes of Health (NIH) | R01DK088423 | Joel K Elmquist |
| National Institutes of Health (NIH) | R37DK053301 | Joel K Elmquist |
| National Institutes of Health (NIH) | R01DK55758 | Philipp E Scherer |
| National Institutes of Health (NIH) | U19DK062434 | David J Mangelsdorf |
| Howard Hughes Medical Institute (HHMI) | | David J Mangelsdorf |
| National Institutes of Health (NIH) | R01DK099110 | Philipp E Scherer |

The funders had no role in study design, data collection and interpretation, or the decision to submit the work for publication.

## Author contributions

VM-A, LG, Conception and design, Acquisition of data, Analysis and interpretation of data, Drafting or revising the article; SL, DJM, JKE, Conception and design, Analysis and interpretation of data, Drafting or revising the article; ALB, KS, YZ, Conception and design, Acquisition of data, Analysis and interpretation of data; CMK, Conception and design, Acquisition of data, Analysis and interpretation of data, Drafting or revising the article, Contributed unpublished essential data or reagents; PES, Conception and design, Drafting or revising the article

## Author ORCIDs

David J Mangelsdorf, http://orcid.org/0000-0002-4355-0796

## Ethics

Animal experimentation: This study was performed in strict accordance with the recommendations in the Guide for the Care and Use of Laboratory Animals of the National Institutes of Health. All of the animals were handled according to approved institutional animal care and use committee (IACUC) protocols (2012-0206) of UTSW. The protocol was approved by the Committee on the Ethics of Animal Experiments of UTSW. Every effort was made to minimize suffering.

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
