## [Decision Letter]

Thank you for sending your work entitled “Loss of LXRα/β in Peripheral Sensory Neurons Modifies Energy Expenditure” for consideration at *eLif*e. Your article has been favorably evaluated by a Senior editor and three reviewers, one of whom is a member of our Board of Reviewing Editors.

The Reviewing editor and the other reviewers discussed their comments before we reached this decision, and the Reviewing editor has assembled the following comments to help you prepare a revised submission.

The reviewers were in agreement that the work was novel and potentially of interest to the readers of *eLife*. All of the reviewers felt that the delineation of a role for sensory nerve metabolism in control of systemic physiology was exciting.

At the same time, the review process identified a few opportunities to strengthen the work.

1) Some of the reviewers felt that the observed effect on browning was modest and that it would be important to consider the potential contribution of muscle to the energy expenditure phenotype. Is there evidence of increased fatty acid uptake and oxidation?

2) The suggestion that LXRs sense changes in fatty acid levels is provocative, but not currently supported by data. The authors could address whether LXR target gene expression in sensory neurons change during fasting/refeeding. Additional discussion of how this might be accomplished would also be useful.

Minor comments:

1) Please confirm that the knockout mice employed have been appropriately backcrossed to a common genetic background. Genetic background and degree of backcrossing should be indicated clearly in Methods.

---

## [Author Response]

*1) Some of the reviewers felt that the observed effect on browning was modest and that it would be important to consider the potential contribution of muscle to the energy expenditure phenotype. Is there evidence of increased fatty acid uptake and oxidation*?

First, as suggested we changed the title to “Loss of the Liver X Receptor LXRα/β in Peripheral Sensory Neurons Modifies Energy Expenditure’’.

The reviewers suggested that we study the potential contribution of muscle to the energy expenditure phenotype. To establish whether peripheral sensory neuron-specific deletion of LXRs alters skeletal muscle energy expenditure, we examined mitochondrial electron transport chain activity by performing mitochondrial electron-flow and electron-coupling experiments to assess oxygen- consumption rates (OCRs), a previously validated method used to quantify mitochondrial oxidative capacity and integrity (Kusminski CM NatMed 2012; Rogers GW PLoS ONE 2011). Interestingly, during the electron-flow analyses, we observed that skeletal muscle mitochondria derived from knockout (LXRs^Nav^) mice exhibit markedly higher oxygen-consumption rates (OCRs) in response to the substrates pyruvate, malate, succinate and ascorbate (revised version Figure 3). Furthermore, electron-coupling experiments to gage mitochondrial coupling and integrity revealed no defects in skeletal muscle mitochondrial function in either genotype (revised Figure 3—figure supplement 1). We thank the reviewers for this excellent suggestion. The data described in the revised article (Results section) indicate that deletion of LXR specifically in neuronal nodose ganglion enhances skeletal muscle mitochondrial oxidative respiration.

*2) The suggestion that LXRs sense changes in fatty acid levels is provocative, but not currently supported by data. The authors could address whether LXR target gene expression in sensory neurons change during fasting/refeeding. Additional discussion of how this might be accomplished would also be useful*.

The reviewers also inquired whether LXR target gene expression in sensory neurons change during fasting/refeeding. As suggested, we used a new cohort of littermate mice to study whether fasting-induced increases in fatty acid availability potentially modified nodose ganglia gene expression. We found that *Abca1*, *Srepb1c* and Synuclein mRNA levels were significantly changed in LXRs^fl/fl^ mice that were fasted for 20 hours (see revised Figure 4). Notably, this increase in fasted *Abca1* and Synuclein was blunted in LXRs^Nav^ mice (revised Figure 4). These in vivo data suggest that nodose ganglia neurons may sense circulated or secreted cues during starvation and respond by regulating unique LXR-dependent genes.

Despite its well-documented role in regulating the transcription of genes crucial for lipid synthesis and storage upon cholesterol sensing, little is known about how LXRs function in peripheral neurons and further investigations will be necessary to completely understand the role of these NRs in the nodose ganglia neurons. Our study suggests that LXRs in vagal sensory neurons potentially regulate vagal synaptic transmission, ultimately affecting the gating of information to adipose tissues and muscle. Since WAT, BAT and muscle receive innervation from sympathetic neurons, we suspect that the increased sympathetic tone—that is secondary to altered input from the parasympathetic vagal sensory neuron activity—may underlie the increased energy expenditure observed in LXRs^Nav^ mice. These results and discussion were added to the revised manuscript (please see the subsection “The loss of LXRα/β in Nav1.8 expressing neurons attenuates lipid accumulation in brown adipose tissue, promotes browning in subcutaneous fat and modifies NG gene expression”). We also added a section detailing the mitochondrial studies to the Methods and made legend modifications.

*Minor comments*:

*1) Please confirm that the knockout mice employed have been appropriately backcrossed to a common genetic background. Genetic background and degree of backcrossing should be indicated clearly in Methods*.

As asked, we have detailed the genetic background of the mice in the Methods section. LXR^fl/fl^ mice (on a mixed C57BL6 and 129SV background) were bred and kept as a mixed background in a closed colony in the UTSW germ-free rodent facility. LXRs^fl/fl^ mice were then backcrossed to C57BL/6J mice for six generations prior to experiments. Transgenic mice (C57/BL6 background) carrying *Cre*-recombinase driven by a Nav1.8 promoter (Nav1.8::Cre) were bred with backcrossed to the LXRs^fl/fl^ mice. LXRs^fl/fl^ mice were then bred with LXRs^fl/fl^ Nav^+/-^ mice to generate at least 3 cohorts of littermate LXRs^Nav^ and LXRs^fl^ mice that were used and compared in each of the respective experiments.